# Co-Encapsulation of Paclitaxel and JQ1 in Zein Nanoparticles as Potential Innovative Nanomedicine

**DOI:** 10.3390/mi13101580

**Published:** 2022-09-22

**Authors:** Marilena Celano, Agnese Gagliardi, Valentina Maggisano, Nicola Ambrosio, Stefania Bulotta, Massimo Fresta, Diego Russo, Donato Cosco

**Affiliations:** Department of Health Sciences, Campus Universitario “S Venuta”, University “Magna Græcia” of Catanzaro, I-88100 Catanzaro, Italy

**Keywords:** JQ1, nanoparticles, paclitaxel, triple-negative breast cancer, zein

## Abstract

The manuscript describes the development of zein nanoparticles containing paclitaxel (PTX) and the bromo-and extra-terminal domain inhibitor (S)-tertbutyl2-(4-(4-chlorophenyl)-2,3,9-trimethyl-6H-thieno(3,2-f)(1,2,4)triazolo(4,3-a)(1,4)diazepin-6-yl)acetate (JQ1) together with their cytotoxicity on triple-negative breast cancer cells. The rationale of this association is that of exploiting different types of cancer cells as targets in order to obtain increased pharmacological activity with respect to that exerted by the single agents. Zein, a protein found in the endosperm of corn, was used as a biomaterial to obtain multidrug carriers characterized by mean sizes of ˂200 nm, a low polydispersity index (0.1–0.2) and a negative surface charge. An entrapment efficiency of ~35% of both the drugs was obtained when 0.3 mg/mL of the active compounds were used during the nanoprecipitation procedure. No adverse phenomena such as sedimentation, macro-aggregation or flocculation occurred when the nanosystems were heated to 37 °C. The multidrug nanoformulation demonstrated significant in vitro cytototoxic activity against MDA-MB-157 and MDA-MB-231 cancer cells by MTT-test and adhesion assay which was stronger than that of the compounds encapsulated as single agents. The results evidence the potential application of zein nanoparticles containing PTX and JQ1 as a novel nanomedicine.

## 1. Introduction

The most common tumor-related disease in women is breast cancer and it is one of the leading causes of death [1]. The expression of the estrogen receptor (ER) and the progesterone receptor (PR), as well as the amplified status of human epidermal growth factor receptor 2 (HER2), are parameters involved in the current classification of breast cancer, and they are also used to identify the most useful therapeutic approaches. In particular, triple-negative breast cancer (TNBC), characterized by the lack of the aforementioned characteristics, affects younger women and carries a poor prognosis. This is due to its frequent tendency to metastasize, and it accounts for 12–17% of all malignant breast tumors [2]. Only early-stage TNBC is chemotherapy-sensitive, but, unfortunately, early diagnosis and treatment are still challenging and the incidence of breast cancer is increasing globally [1,2].

Specific chemotherapy for the treatment of TNBC is currently unavailable [3,4]. Paclitaxel (PTX), a lipophilic chemotherapeutic agent belonging to the taxane family, is an efficacious anticancer agent and is also being used against TNBC in clinical practice [3,5]. Unfortunately, allergic reactions, myelosuppression, and digestive tract reactions are the highly undesirable side effects of PTX, most of them related to the organic solvents used to solubilize the active compound, a fact which has widely compromised its clinical application [4,6]. For this reason, innovative strategies are being employed in order to avoid these and nanomedicine has emerged as an efficacious approach in cancer chemotherapy [7,8]. In fact, the delivery of the active compounds directly into tumors by means of specific nanocarriers promotes the decrease in their efficacious dosage and the related side effects [9,10]. The use of albumin nanoparticles containing PTX is a well-known example of a nanomedicine that preserves the pharmacological efficacy of the lipophilic drug, while at the same time avoiding the use of toxic organic solvents [11,12].

The association of PTX with other classes of active compounds represents another therapeutic option for the treatment of TNBC [13,14]. During carcinogenesis, extensive epigenetic modifications occur, resulting in dysregulated gene expression and abnormal proliferation. This means that molecules able to target the epigenome, in particular the bromodomain and the extraterminal protein (BET) signaling pathway that plays such an essential role in cell proliferation, immunitary responses, and pro-inflammatory events, are all being evaluated in numerous experimental investigations [15,16,17,18,19]. Among the BET inhibitors, JQ1 has a strong antiproliferative effect against TNBC cell lines and xenografts [15,17]. In fact, our research team recently encapsulated the lipophilic molecule within PLGA nanoparticles, obtaining an increase in its half-life and preserving its anticancer activity [20].

Recent studies demonstrated that the association of BET inhibitors and other drugs have favorable therapeutic outcomes [17,21]. Zhou et al. demonstrated the rationale for using a combination of BET inhibitors and traditional chemotherapy in the treatment of non-small cell lung carcinoma [22]. Moreover, it was shown that BET inhibition sensitized breast tumors to chemotherapy, hormone therapy, and PI3K inhibitors in vitro [17].

The concept of the co-encapsulation of two or more active compounds within a drug delivery system has been extensively investigated over the last few decades in order to increase the therapeutic efficacy of various drugs and give good results [23,24]. As a successful example, the liposomal formulation Vyxeos can be cited since it was recently approved for clinical use on humans [22,23,24,25]. Several colloidal systems have been developed and characterized with the aim of evaluating useful associations of molecules within specific biocompatible carriers, but a relatively scarce number of formulations has actually reached the market [26,27]. Besides the various biocompatible nanocarriers proposed for pharmaceutical application, our research team recently focused on the development of vegetal protein-based systems. This is because these biomaterials are characterized by peculiar physico-chemical properties useful for obtaining colloidal structures, are readily available at low cost, and are also considered “environmentally economic” as compared to animal proteins [28,29,30,31]. In specific, zein, a highly hydrophobic protein with a peculiar bricklike shape, was employed to obtain biodegradable colloidal systems able to retain several active compounds [32,33,34,35,36,37,38,39].

In particular, zein nanoparticles have been widely exploited in the last few years in order to improve the bioavailability of several antitumor drugs [40,41,42]; among these, PTX was efficaciously entrapped in the protein structure, obtaining a formulation characterized by significant stability in polar media and the ability to increase the cytotoxicity of the active compound against MCF-7 and k562 cells [43]. Considering the results, the idea of this investigation was to develop a zein-based multidrug nanocarrier containing JQ1 and PTX with the aim of obtaining a novel potential nanomedicine to be used for the treatment of TNBC.

## 2. Materials and Methods

### 2.1. Materials

Zein, sodium deoxycholate monohydrate (SD), paclitaxel (PTX), 3-[4,5-dimethylthiazol-2-yl]-3,5-diphenyltetrazolium bromide salt (used for MTT-tests), phosphate buffered saline (PBS) tablets, dimethyl sulfoxide, and amphothericin B solution (250 μg/mL) were all purchased from Sigma Aldrich (Milan, Italy). The bromo- and extra-terminal domain (BET) inhibitor (S)-tertbutyl2-(4-(4-chlorophenyl)-2,3,9-trimethyl6H-thieno(3,2-f)(1,2,4)triazolo(4,3-a)(1,4)diazepin-6-yl)acetate (JQ1), DMEM (Dulbecco’s Modified Eagle’s Medium), RPMI, tripsin/EDTA, penicillin/streptomycin solution, and fetal bovine serum (FBS) were purchased from Thermo Fischer Scientific Inc. (Waltham, MA, USA). Human breast cancer cells (MDA-MB-157 and MDA-MB-231) were purchased from the American Type Culture Collection (Manassas, VI, USA). All other materials and solvents used in this investigation were of analytical grade (Carlo Erba, Milan, Italy).

### 2.2. Preparation of Zein Nanoparticles

The zein nanoparticles were prepared according to the nanoprecipitation method of the pre-formed polymer in an aqueous solution as previously described [43]. Briefly, the vegetal protein (3.3 mg/mL) was dissolved in an ethanol/water solution (3 mL, 2:1 *v/v*) at room temperature (20 °C) and successively added to an aqueous phase (5 mL) enriched with SD (1.25% *w/v*), homogenized at 24,000 rpm for 1 min (Ultraturrax^®^ model T25, IKA^®^ Werke, Wilmington, NC, USA), and then mechanically stirred on a magnetic plate (600 rpm for 12 h at 20 °C) to favor the evaporation of the organic solvent. The final protein concentration was 2 mg/mL.

Zein nanoparticles containing JQ1 or PTX or their association were obtained by adding different amounts of the active compound(s) to the organic phase (Table 1). The resulting formulations were centrifuged at 90 k rpm for 1 h using a Beckman Optima™ Ultracentrifuge (Fullerton, NU, Canada) in order to perform the analytical investigations, or suitably purified by means of Amicon^®^ Ultra centrifugal filters (Sigma Aldrich, Saint Louis, MO, USA) (cut-off 10 kDa, 4000 rpm for 120 min) before the cytotoxicity experiments.

### 2.3. Characterization of Zein Nanoparticles

The mean diameter, size distribution and surface charges of the nanosystems were evaluated by means of a Zetasizer NanoZS (Malvern Panalytical Ltd., Spectris plc, Malvern, UK) [34]. The results are the mean of three different measurements performed in triplicate on three different samples (10 determinations for each sample) ± standard deviation and expressed as a function of the intensity parameter. The morphology of the nanosystems was investigated by Transmission Electron Microscopy (TEM) as previously described [44].

The formulations were also submitted to Turbiscan Lab^®^ Expert analysis (Formulaction, Toulouse, France) in order to evaluate their stability as a function of temperature and storage time [45]. The results were reported as Turbiscan Stability Index (TSI) versus time.

The entrapment efficiency of PTX and JQ1 was investigated previously and described by our research team [20,43].

### 2.4. TNBC Cell Lines and Proliferation Assay

Human breast cancer epithelial cells, MDA-MB-231 and MDA-MB-157, widely used as models of TNBC for their characteristics of growth and progression, were cultured in DMEM or RPMI 1640 (Thermo Fisher Scientific Inc., Waltham, MA, USA) and maintained at 37 °C in humidified 5% CO_2_ [46].

Cell proliferation was analyzed by MTT assay after 24, 48, and 72 h of incubation with JQ1, PTX, and both compounds encapsulated in the nanoparticles [47]. MDA-MB-231 and MDA-MB-157 cells were seeded in 96-well plates at a concentration of 5 × 10^3^ and 3.5 × 10^3^/well, respectively. After treatment with JQ1 (3 nM or 60 nM) and PTX (1.5 nM or 30 nM) in the free form, entrapped within the zein nanoparticles as single agents (nPTX or nJQ1) or as multidrug carriers (nPTX-JQ1) the crystals of formazan were quantified using a microplate spectrophotometer (Thermo Fisher multiskan FC) at a wavelength of 540 nm and a reference wavelength of 690 nm. The results are expressed as percentages over untreated cells and cells treated with empty nanoparticles.

### 2.5. Adhesion Assay

Adhesion assays were performed as previously described [48]. MDA-MB-231 and then the MDA-MB-157 cells were seeded in 6-well plates (1.6 × 10^5^ or 1.3 × 10^5^ for each well) and incubated with JQ1 (3 nM), PTX (1.5 nM), empty nanoparticles, nPTX, nJQ1, or nPTX-JQ1 (at the same concentrations used for the free forms of the drugs). After forty-eight hours, 5 × 10^4^ cells were plated into 24-well plates coated with collagen I (BD Biosciences, Milan, Italy) and, after 30 min of incubation, the cells were stained with a 0.1% Crystal Violet solution solubilized in 10% acetic acid. Cell attachment was quantified by measuring the absorbance at 560 nm with a Thermo Fisher multiskan FC spectrophotometer. Results are expressed as percentages over untreated cells and cells treated with unloaded nanoparticles are indicated as ‘Control’.

### 2.6. Statistical Analysis

The results of the proliferation and adhesion assays were analyzed by a one-way ANOVA followed by the Tukey–Kramer multiple comparison test using the GraphPad Prism version 5.0 statistical software (GraphPad Software Inc., San Diego, CA, USA). The results are expressed as means ± standard deviation (SD) and *p* values under 0.05 were considered statistically significant.

## 3. Results and Discussion

### 3.1. Characterization of Zein Nanoparticles

After several years of work, our research team developed a colloidal formulation made up of zein as the main component and the anionic surfactant SD as a stabilizer and proposes these nanosystems as potential drug carriers for IV administration as a consequence of their significant physical stability and compatibility with the physiological medium [34]. PTX is one of various active compounds entrapped within zein nanoparticles and nanoencapsulation did not compromise the pharmacological activity of the drug [43]. The ideal concentration selected to prepare the nanosystems was 0.3 mg/mL of the lipophilic compound because several destabilization phenomena occurred (sediment and macroaggregates) when this amount was exceeded (Table 1) [43]. The first step of this investigation was focused on the evaluation of the physico-chemical parameters of the nanosystems containing the BET inhibitor JQ1, in order to determine the maximum amount of the compound that can be used during the preparation procedure, mandatory information required to develop the multidrug systems. The encapsulation of JQ1 promoted an increase in the mean diameter and polydispersity index of the zein nanoparticles which was proportional to the concentration of the added drug (Table 1). Specifically, any amount of JQ1 over 0.5 mg/mL induced a negative impact on the size distribution of the colloidal systems while a concentration equal to 0.7 mg/mL promoted the formation of macroaggregates and sediments (data not shown). Contrarily, the surface charge of the zein nanoparticles was not affected by the lipophilic compound and a Zeta-potential value of ~−30 mV was obtained in all samples. Considering these results, 0.3 and 0.5 mg/mL of JQ1 were associated to various amounts of PTX (0.1–0.3 mg/mL) during the preparation of the zein nanoparticles in order to evaluate the best concentrations to be used as a function of the aforementioned parameters (Table 1). Table 1 shows the detrimental effect on the particle sizing exerted by the highest concentration of JQ1 when it was co-encapsulated with PTX. In fact, the appearance of aggregates resulted when even as little as 0.1 mg/mL of the taxane derivative was used, while the association of 0.3 mg/mL of JQ1 to PTX favored the development of nanoparticles characterized by physico-chemical parameters compatible with systemic administration (Table 1).

TEM analysis demonstrated a spherical morphology of the nanosystems, substantiating the results already published by our research team [49]. It also confirmed the absence of aggregates when the two active compounds were co-encapsulated within the protein matrix (Figure 1).

The physical stability of the zein nanoparticles was investigated by means of a Turbiscan Lab apparatus in order to evaluate up to what point the entrapped compounds can cause potential adverse phenomena. The addition of PTX during the preparation procedure did not dramatically modify the TSI profile (the drug concentration of 0.3 mg/mL induced the marked variation) of the zein nanosystems as previously described and a temperature of 37 °C exerted a positive effect as a consequence of the rearrangement of the colloidal structure [43] (Figure 2). This trend is peculiar to zein-based systems and was demonstrated in several experimental investigations [37,38]. The encapsulation of JQ1 had similar effects; namely, drug concentrations over 0.5 mg/mL promoted a noticeable increase in the TSI profile, confirming the photo-correlation spectroscopy results already discussed. In this case, too, the temperature promoted a stabilization of the colloidal structure as demonstrated by the decrease in the TSI slope (Figure 2). It was interesting to observe the TSI profiles of the zein nanoparticles prepared with 0.3 mg/mL of the two drugs; namely, the curve slope fell between that obtained following the encapsulation of the active compounds as a single agent, demonstrating that a positive effect is exerted by JQ1 on the colloidal structure when large amounts of PTX are used (Figure 2). Specific analytical experiments on the potential chemical interaction between the two molecules and the protein are in progress in order to evaluate precisely this topic. The described results confirmed that it is possible to entrap the two drugs within zein nanoparticles without provoking adverse phenomena such as sedimentation, macro-aggregation or flocculation when 0.3 mg of compound are used during the sample preparation.

### 3.2. Evaluation of the Drug Entrapment Efficiency

The evaluation of the amount of a drug retained by a colloidal system must be investigated during the phase of characterization of the formulation in order to identify the concentration of the active compound that remains associated to the carrier subsequent to the purification step. In the case of a multidrug formulation, this information is required so that the ideal amount of drug to be used during the sample preparation can be determined in order to obtain a good degree of retention in a delivery system. Our research team already demonstrated that PTX can be encapsulated within zein nanoparticles, obtaining an entrapment percentage of about 40% of the active compound in the colloidal matrix when 0.3 mg/mL of drug is used (Table 2) [43]. Similar results were obtained with the use of JQ1 and more than 50% of the compound was retained when 0.5 and 0.6 mg/mL of drug were used initially (Table 2). As previously noted, though, these concentrations promote a significant increase in the mean diameter of the particles and so are detrimental to their physical stability. For this reason, the evaluation of the entrapment efficiency was performed on a multidrug formulation prepared with 0.3 mg/mL of JQ1 and varying amounts of PTX. It was interesting to note that the use of 0.3 mg/mL of both active compounds gave a similar retention of the two drugs (~35%) and this association allowed us to obtain a formulation characterized by a molar concentration of JQ1 that was 2-fold higher than that of PTX as a consequence of their molecular weights.

The use of a greater amount of JQ1 in association with PTX had a negative effect on the entrapment efficiency of the two molecules, probably as a consequence of the destabilization of the protein structure, as can be seen in Section 3.1 (Table 2).

Considering the results, the multidrug formulation obtained by the association of 0.3 mg/mL of the two active compounds was used in the cytotoxicity experiments.

### 3.3. Evaluation of the Cytotoxicity and Effects on the Adhesion of TNBC Cells

Initially, the effects of JQ1 and PTX as single compounds and in association were evaluated on MDA-MB-231 and in MDA-MB-157 cells. As shown in Figure 3, a concentration- and time-dependent decrease in proliferation was observed in both cell lines, having similar profiles (Figure 3). A strong inhibitory effect was observed after 72 h incubation when the highest concentrations of the drugs were used together (JQ1 60 nM with PTX 30 nM ~70% vs. controls, *p* < 0.001) (Figure 3A). The nanoencapsulation of the two active compounds within the zein nanoparticles increased their cytotoxicity and the multidrug formulation provided the best results (Figure 3B). In particular, an inhibition of 75% of cell viability was observed on the MDA-MB-231 cells when they were incubated with JQ1-loaded zein nanoparticles (60 nM) or PTX-loaded protein nanosystems (30 nM), while the multidrug formulation promoted the strongest cytotoxic effects on both cell lines, inducing an almost complete inhibition of their viability when the highest drug concentration was used (~90% over control, *p* < 0.001) (Figure 3B). Similar results were observed for the MDA-MB-157 cells (Figure 4).

Our research team showed that the time-dependent uptake of zein nanoparticles in MCF-7 and K562 cells together with this peculiar feature could promote a better pharmacological action of the entrapped compounds as a consequence of their massive localization in the cytosol [43].

In this investigation, the effects of the various formulations on the cell adhesion properties were also evaluated at the lowest drug concentration. In particular, a slight reduction in adhesion of MDA-MB-231 cells was observed (~30% vs. control) when they were tested in the free form or encapsulated in zein nanoparticles as single agents (Figure 5). The co-encapsulation of the two active compounds in the colloidal systems exerted a stronger effect and also gave a better reduction in cell adhesion (~55% vs. control) (Figure 4). Similar results were obtained on MDA-MB-157 cells as well, confirming the trend of cytotoxicity previously described (data not shown). Again, in both cells, the effects of nPTX-JQ1 (containing PTX 1.5 nM and JQ1 3 nM, respectively) were stronger than those of the compounds encapsulated as single agents (Figure 5).

## 4. Conclusions

Currently, the management of TNBC that is unresponsive to the current treatment approaches is still a main clinical challenge and for this reason it is important to develop novel therapeutic strategies [3,50,51,52]. In this study, we demonstrated that zein nanoparticles can be used as biocompatible carriers for the co-delivery of the BET inhibitor JQ1 and PTX, two molecules characterized by a strong lipophilic character [4,53]. The zein nanoparticles favored the administration of the two active compounds in the polar media and they did not compromise the cytotoxic activity of the entrapped drugs. Moreover, it was observed that the association of JQ1 and PTX within the zein-based nanosystems exerted increased pharmacological activity on the TNBC cells. Additional investigation will be required in order to evaluate (i) the chemical interaction between the various components of the multidrug formulation, (ii) the synergistic cytotoxic activity between the two drugs and (iii) the real efficacy of the herein described nanomedicine by preclinical in vivo studies.

## Figures and Tables

**Figure 1 micromachines-13-01580-f001:**
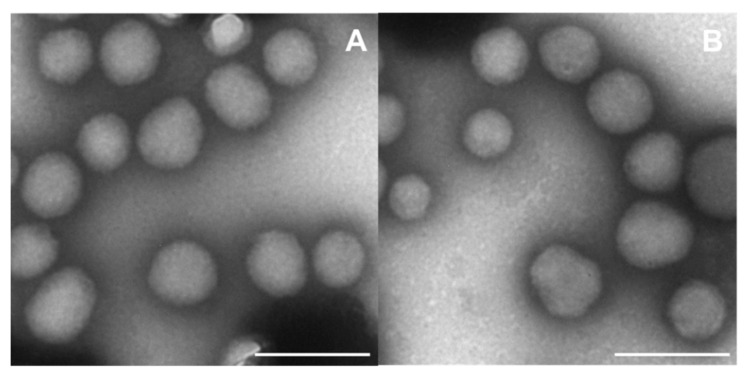
TEM analysis of (**A**) empty zein nanoparticles (2 mg/mL of protein stabilized with sodium deoxycholate, (**B**) nanosystems prepared with 0.3 mg/mL of JQ1 and PTX. Magnification: 30,000×, Bar = 200 nm.

**Figure 2 micromachines-13-01580-f002:**
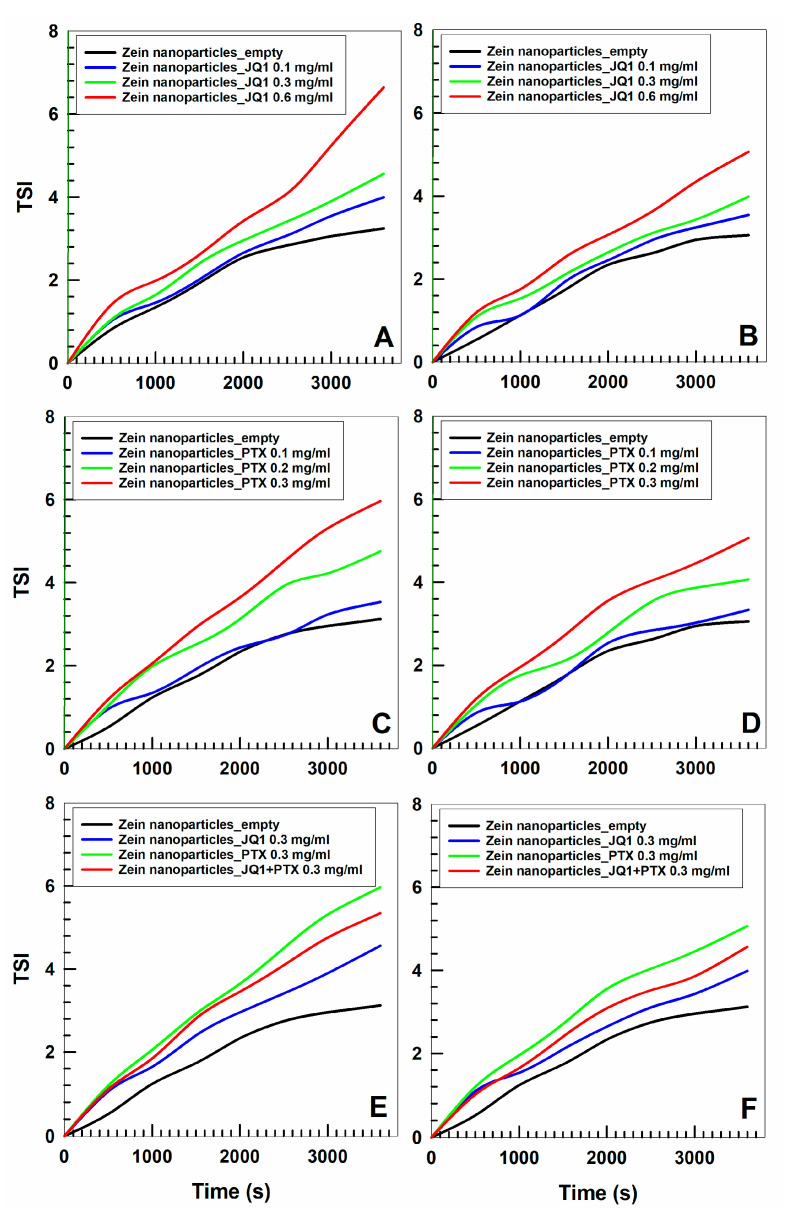
TSI profile of zein nanoparticles prepared with various amounts of JQ1 (0.1, 0.3, 0.6 mg/mL) and PTX (0.1, 0.2, 0.3 mg/mL) as a single agent or as multidrug carrier (0.3 mg/mL of each active compound) as a function of time and temperature (panels (**A**,**C**,**E**): 20 °C; panels (**B**,**D**,**F**): 37 °C).

**Figure 3 micromachines-13-01580-f003:**
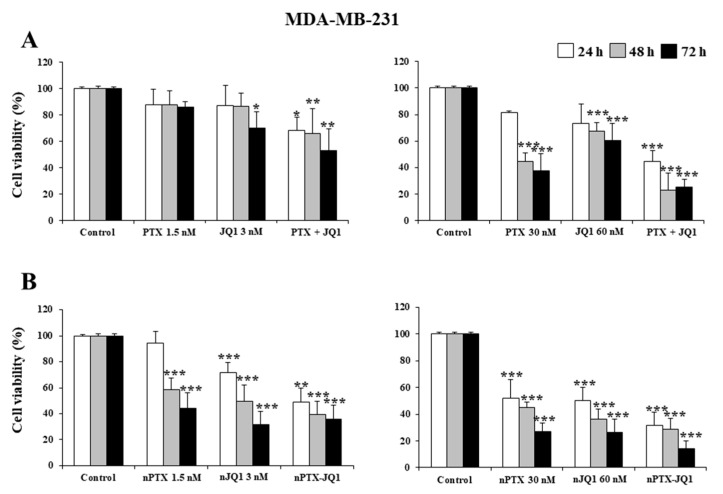
Effects of JQ1 and PTX on TNBC cell viability. MDA-MB-231 cells were treated with JQ1 and PTX as a free form (**A**) or encapsulated in zein nanoparticles (**B**) as single agents (nJQ1, nPTX) or as multidrug carriers (nJQ1-PTX). Each experiment was performed in triplicate and values are expressed in % over untreated cells (control), as mean ± SD. Statistical analysis was performed using the Tukey–Kramer multiple comparisons test. * *p* < 0.05, ** *p* < 0.01, *** *p* < 0.001 vs. control.

**Figure 4 micromachines-13-01580-f004:**
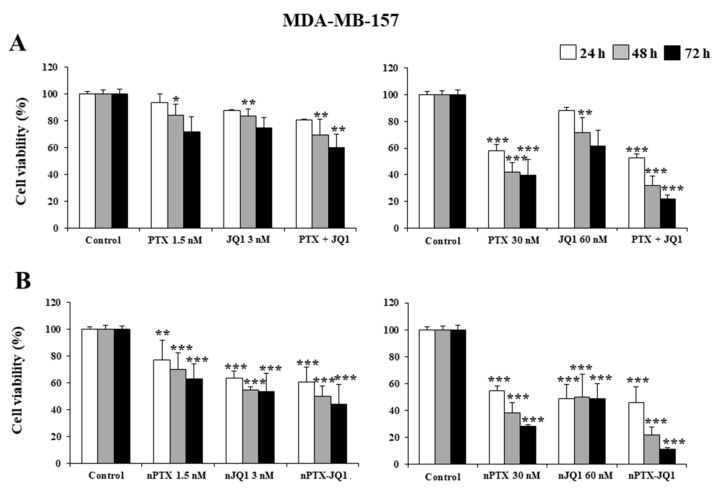
Effects of JQ1 and PTX on TNBC cell viability. MDA-MB-157 cells were treated with JQ1 and PTX in the free form (**A**) or encapsulated in zein nanoparticles (**B**) as single agents (nJQ1, nPTX) or as multidrug carriers (nJQ1-PTX). Each experiment was performed in triplicate and values are expressed in % over untreated cells (control), as mean ± SD. Statistical analysis was performed using the Tukey–Kramer multiple comparisons test. * *p* < 0.05, ** *p* < 0.01, *** *p*< 0.001 vs. control.

**Figure 5 micromachines-13-01580-f005:**
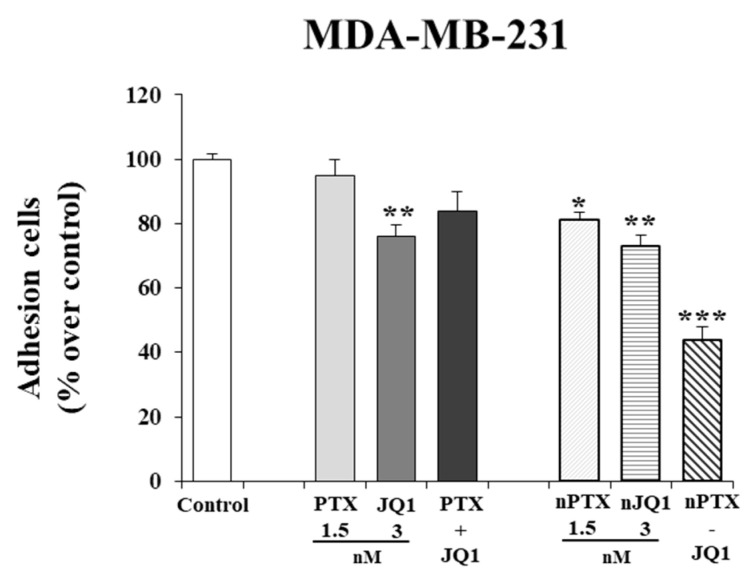
Effects of JQ1 and PTX on TNBC adhesion properties. MDA-MB-231 cells were prepared for adhesion assays as indicated in the materials and methods section. Each experiment was performed in triplicate and values are expressed in % over untreated cells (control), as mean ± SD. Statistical analysis was performed using the one-way ANOVA test. * *p*< 0.05, ** *p*< 0.01, *** *p*< 0.001 *vs* control.

**Table 1 micromachines-13-01580-t001:** Physico-chemical properties of PTX-loaded SD-stabilized zein nanoparticles.

JQ1 (mg/mL)	PTX (mg/mL)	Mean Sizes (nm)	Polydispersity Index	Zeta Potential (mV)
-	-	110 ± 2	0.11 ± 0.01	−32 ± 1
0.1	-	112 ± 1	0.12 ± 0.01	−30 ± 2
0.2	-	119 ± 2	0.14 ± 0.02	−30 ± 1
0.3	-	125 ± 1	0.15 ± 0.03	−28 ± 2
0.4	-	152 ± 3 **	0.19 ± 0.03	−32 ±3
0.5	-	161 ± 3 **	0.20 ± 0.03	−31 ± 1
0.6	-	220 ± 3 **	0.27 ± 0.05	−29 ± 2
-	0.1	103 ± 6	0.19 ± 0.01	−32 ± 1
-	0.2	131 ± 6 **	0.18 ± 0.08	−28 ± 2
-	0.3	158 ± 5 **	0.21 ± 0.04	−31 ± 2
	0.4	515 ± 14 **	0.64 ± 0.10	−35 ± 5
0.3	0.1	128 ± 2 *	0.18 ± 0.01	−30 ± 2
0.3	0.2	144 ± 3 *	0.17 ± 0.01	−29 ±3
0.3	0.3	161 ± 2 **	0.19 ± 0.02	−29 ± 2
0.5	0.1	199 ± 3 **	0.24 ± 0.03	−38 ±2
0.5	0.2	220 ± 3 **	0.25 ± 0.03	−34 ± 3
0.5	0.3	277 ± 4 **	0.32 ±0.04	−38 ± 3

* *p* ˂ 0.05, ** *p* ˂ 0.001 (with respect to the empty formulation).

**Table 2 micromachines-13-01580-t002:** Entrapment efficiency of JQ1 and PTX within zein nanoparticles as a function of the drug concentration.

JQ1 (mg/mL)	PTX (mg/mL)	EE	
		JQ1 (%)	PTX (%)
0.1	-	33 ± 4	-
0.2	-	41 ± 3	-
0.3	-	44 ± 2	-
0.4	-	48 ± 3	-
0.5	-	59 ± 4	-
0.6	-	52 ± 5	-
-	0.1	-	30 ± 3
-	0.2	-	33 ± 4
-	0.3	-	42 ± 3
0.3	0.1	40 ± 3	25 ± 3
	0.2	36 ± 4	29 ± 3
	0.3	35 ± 3	33 ± 4
0.5	0.1	19 ± 4	21 ± 2
	0.2	23 ± 5	24 ± 3
	0.3	28 ±4	25 ± 4

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
