# Peer review of "Co-Encapsulation of Paclitaxel and JQ1 in Zein Nanoparticles as Potential Innovative Nanomedicine"

_micromachines, 2022, doi:10.3390/mi13101580_

Round 1

Reviewer 1 Report

In this study, the authors developed biopolymer nanoparticles encapsualting 2 drugs for cancer therapy. The manuscript entitled “Co-encapsulation of paclitaxel and JQ1 in zein nanoparticles as potential innovative nanomedicine". The results presented in this manuscript are significant.  I recommend it for acceptance with very minor revision.

Page 2 :  NSCLC abbreviation is not identified in text

Page 3 in Zein preparation section: the authors should clarify the followings

SD  first appearence of this abbreviation

Method of preparation requires more details. Stirring rate and time are missing 

Homogenization rate is missing 

Model of equipment are missing in this part 

Volume of organic phase used and what was its composition?

Method of analysis for the 2 drugs simultaneously is not mentiod for EE%

In Results and Discussion part:  Did the authors examined other surfactants  for the development of formulae.?  Clarify 

Figure one lack magnification value

Table one better to be shifted in results and discussion part and not in expirmental 

Reviewer 2 Report

1. Page 3: 2.2, room temperature: Please specify the temperature

2. Page 3: 2.2, evaporation, specify condition of temp and pressure (if any)

3. Is protein precipitation is issue with ethanoic extract or temperature effect? My concern is stability of protein in regards to process used in 2.2.

4. In 2.2 process, is there any stabilizers/surfactant used? If yes mentioned it, if no then how particle aggregation was avoided.

5. Figure 1: Quality is poor, High resolution with visible scale is preferred

6. Cut short conclusion (Presently its summary form)

Reviewer 3 Report

The current work by Cosco et al. described the co-encapsulation of paclitaxel and JQ1 in zein nanoparticles for potential breast cancer treatment. I have the following comments and suggestions

1, The authors applied sodium deoxycholate as stabilizing agents. If I am not wrong it is usually used in lysis buffer. Can the authors clarify whether SD can be fully removed during the production procedure? If it is fully removed, can the NPs still be stable within aqueous solution?

2, Could the authors further clarify extra information regarding to the combination of the drugs? For example, the authors could calculate the Combination Index (CI), in particular this dose–effect-based approaches rely on Loewe Additivity model. Meanwhile, the benefit of a combination therapy is not simply due to the property of the drugs, but could also depend on the dose ratio. Therefore a multiple-ray design exploring a given set of fixed ratios (the dose of one drug is escalated while the dose of the second remains constant) should be conducted, so the authors should be able to optimize the loading degree of both drugs and confirm that at this concentration one can achieve the best CI.

3, I suggest the authors changing the title of y-axis of Figure 3-4 from absorbance into Viability (%).

Reviewer 4 Report

The manuscript entitled "Co-encapsulation of paclitaxel and JQ1 in zein nanoparticles as potential innovative nanomedicine" described the encapsulation of paclitaxel and JQ1 in zein nanoparticles to be investigated for their effect on cancer cells.
The manuscript is interesting and can be accepted for publications after major corrections:
1- The abstract should be modified to give more digital results rather than elastic sentences.
2- First of all the manuscript should be checked by an English native speaker to remove the syntax and typos
3- Page 2 “For this reason, innovative strategies are being employed in order to avoid these and nano-medicine has emerged as an efficacious approach in cancer chemotherapy.
This is because it can deliver the active compounds directly into tumors, decreasing the toxic side effects of the drugs” please consider rephrasing
4- Authors should cite more recent work of zein nanoparticles encapsulating anticancer drugs
5- In the preparation of zein nanoparticles, what is SD?, Please provide the full name before any abbreviation
6- The manuscript describes the combination of two active ingredients for the first time, how did the authors ensure the non-interference between the two drugs during the entrapment efficiency determination?, in addition authors should use prove the chemical interactions. Crystallographic or spectroscopic methods could be performed. At least, FTIR analysis and its interpretation should be given.
7- “The ideal concentration selected to prepare the nanosystems was 0.3 mg/ml of the lipophilic compound because several destabilization phenomena occurred when this amount was exceeded (Table 1)” Table 1 do not show any destabilization effect, the authors should state the parameters measured to investigate the instability of the formed nanoparticles.
8- The authors should clarify the JQ1 concentration investigated with PTX on figure 2
9- The authors should state the IC50 of each drug and the effect of this combination on their IC50

Round 2

Reviewer 4 Report

The authors have answered all the points and the manuscript can be accepted